# Computational modelling of the equine arteritis virus GP5/M Dimer: Implications for immune evasion and virulence

Michael Veit[ID]1*, Anna Karolina Matczuk2

1 Institut für Virologie, Veterinärmedizin, Freie Universität Berlin, Berlin, Germany, 2 Division of Microbiology, Department of Pathology, Faculty of Veterinary Medicine, Wrocław University of Environmental and Life Sciences, Wrocław, Poland

* michael.veit@fu-berlin.de

## Abstract

Equine arteritis virus (EAV) is a positive-stranded RNA virus of the Arteriviridae family. Its GP5/M dimer, the principal component of the viral envelope, mediates virus budding and serves as a key target for neutralizing antibodies. Using Alpha-Fold3, we predicted the 3D structure of the EAV GP5/M dimer and compared it to its homolog in porcine reproductive and respiratory syndrome virus (PRRSV). Both complexes share a conserved architecture comprising a short ectodomain, three helical transmembrane regions, and a β-sheet-rich endodomain. EAV GP5 features a longer ectodomain with four α-helices and a disulfide-linked β-sheet, which forms the most variable and surface-exposed region containing neutralizing epitopes. Adjacent conserved and variable N-glycosylation sites suggest immune evasion mechanisms involving antigenic drift and glycan shielding. Another epitope, located in a membrane-proximal helix, overlaps with known virulence and persistence determinants. The transmembrane domains are the most structurally conserved regions between EAV and PRRSV, characterized by tilted and kinked helices stabilized by hydrophilic interactions within the lipid bilayer. These findings provide molecular insights into the structural organization, immune targets, and virulence-associated features of the GP5/M dimer, offering a foundation for rational vaccine design against EAV.

## Introduction

*Alphaarterivirus equid*, common name equine arteritis virus (EAV), which causes respiratory and reproductive disease in horses and donkeys, is an enveloped plus-strand RNA virus in the Arteriviridae family, which comprises 6 subfamilies [1]. The two species of porcine reproductive and respiratory syndrome virus (PRRSV) are arguably the most relevant viral pathogen in pigs worldwide. PRRSV infection causes abortion and stillbirth in pregnant sows as well as respiratory disease and poor growth performance in piglets. Other relevant arteriviruses include lactate

**Data availability statement:** All data are available in the manuscript, including the Supplementary Files. The PDB-file of the GP5/M model was deposited on 15th October 2025 in ModelArchive (modelarchive.org) with the accession code https://doi.org/10.5452/ma-2p06m.

**Funding:** This study was supported by the German Research Foundation (DFG) in the form of a grant awarded to MV (Grant Number: VE 141/20-1). The publication of this article was funded by Freie Universität Berlin. The specific roles of this author are articulated in the 'author contributions' section. The funders had no role in study design, data collection and analysis, decision to publish, or preparation of the manuscript.

**Competing interests:** The authors have declared that no competing interests exist.

dehydrogenase–elevating virus (LDV) and many species of subfamily *Simarterivirinae* including simian hemorrhagic fever virus (SHFV), which infect non-humane primates and possess zoonotic potential [2–5]. Due to similarities in the genome organization and replication strategy, *Arteriviridae* are grouped in the order *Nidovirales* together with the family *Coronaviridae,* including SARS-CoV-2*.* Consequently, they share proteins with similar structure and function [2,6–9].

EAV is responsible for abortions in pregnant mares, respiratory illness that can be fatal in young foals, and the potential establishment of a persistent carrier state in stallions [10]. Significant Economic losses in the equine industry arise from the multifaceted impact of EAV on reproduction, young horses and breeding operations and cost associated with surveillance, testing and animal movement and trade, which are difficult to estimate [10]. The ability of EAV strains to produce disease can vary greatly, ranging from clinically unapparent to severe. Transmission occurs primarily via the respiratory route and through semen from chronically infected stallions [11].

In Europe, only one inactivated vaccine is currently licensed for EAV, while in the United States, a live-attenuated vaccine is also available—although it is contraindicated in pregnant mares. Although the current vaccines are considered safe and efficacious, immunization against EAV remains infrequent in practice. This is largely due to concerns that vaccine-induced seropositivity may hinder international trade, as there is no DIVA (Differentiating Infected from Vaccinated Animals) or marker vaccine available to distinguish naturally infected animals from vaccinated ones. Although stallions are routinely screened prior to breeding and PCR testing is commonly performed before equine competitions, outbreaks of EAV continue to occur, but are probably underreported [12–16].

Arteriviruses are composed of a nucleocapsid with an asymmetric, linear organization, comprising the viral RNA wrapped by the nucleocapsid protein N, and six membrane proteins, the disulfide-linked GP5/M dimer and the GP2/3/4 complex, the small and hydrophobic E-protein and the ORF5a protein [17,18]. From reverse genetics experiments it is known that arterivirus structural proteins are essential for virus replication, but act at different steps of the replication cycle. If the expression of either GP2 or GP3 or GP4 is abrogated, virus-like particles bud from cells, but the particles are not infectious, indicating that cell entry is disturbed in the absence of the minor glycoprotein complex. In contrast, if either GP5 or M is deleted from the viral genome, no virus particles are released from transfected cells. Thus, GP5 and M are required for virus budding, which does not exclude the possibility that they may have additional functions during virus entry [19,20]. However, the GP2/3/4 complex governs cell tropism, at least in cell culture: When the ectodomains of GP2/GP3/GP4 of EAV and PRRSV are swapped, the cell tropism of the resulting recombinant virus is altered, but not by exchanging the genes encoding GP5 and M [21–23].

Recent genome-wide CRISPR knockout screening identified CD81—a member of the tetraspanin family—as a critical entry receptor for EAV [24]. CD81-expressing cell lines supported EAV entry and replication and a key extracellular loop of the tetraspanin was identified in mediating this interaction. However, additional validation is necessary to confirm CD81 as a bona fide entry receptor, particularly given that the

specific viral structural proteins interacting with CD81 remain unidentified. In parallel, the neonatal Fc receptor (FcRn) has emerged as a significant pro-viral host factor facilitating the entry of several genetically divergent arteriviruses [25]. In the case of PRRSV-2, FcRn has been shown to interact with the viral M and N proteins [26]. Whether FcRn plays a similar role in EAV entry, however, remains to be elucidated. Likewise, which of the viral membrane proteins mediates membrane fusion has not been identified so far for any of the Arteriviruses. Possibly, a complex interplay between minor and major viral membrane proteins determines cell entry of Arteriviruses [27].

M, the most conserved membrane protein of the Arteriviruses consists of a short ectodomain, three putative transmembrane regions, and a long, hydrophilic cytoplasmic tail. M does not have a cleavable signal peptide and is not glycosylated. During Gp5/M heterodimer formation, most M remains as ER-localized monomers that are association-competent and are slowly recruited by newly synthesized GP5 [28,29].

GP5, the major glycoprotein of Arteriviruses, exhibits basically the same sequence of hydrophilic and hydrophobic domains as M. GP5 is targeted to the rough endoplasmic reticulum (ER) by an N-terminal signal peptide and then translocated into the ER lumen where asparagine residues in the sequence context N-X-S/T are modified with carbohydrates and the signal peptide is cleaved. Signal peptides, typically ~30 residues in length, consist of an N-region containing positively charged residues, a hydrophobic H-region and a small C-region with the cleavage site. Whether and where a signal peptide is cleaved by the signal peptidase depends primarily on the presence of small and neutral amino acids (Ala, Gly, Ser, Thr, Cys) at the −1 and −3 positions with respect to the cleavage site. GP5 from distinct PRRSV strains can be cleaved at various sites, and GP5 from a single strain can be cleaved at two separate sites, according to experimental and bioinformatic data [30–32].

Another co-translational modification occurring in the lumen of the ER is the formation of a disulfide-bond between GP5 and M. GP5 of EAV contains five cysteines in its ectodomain, the disulfide-linkage occurs through the N-terminal cysteine. Heterodimerization of M with GP5 is required for their transport from the ER to the Golgi apparatus where they are retained by unidentified signals [33]. GP5 of PRRSV forms a disulfide-linked complex with M involving the only cysteines in their extracellular domains [18]. GP5 and M of both PRRSV-1 and PRRSV-2 are palmitoylated at three and two conserved cysteines, respectively, in close proximity to the transmembrane span. This modification is essential for virus replication, affecting the assembly and budding of virus particles [34].

Several epitopes for neutralizing antibodies have been identified in the ectodomain of GP5 [35–40]. This contrasts with PRRSV, in which the GP2/3/4 complex also serves as a target of neutralizing antibodies [41]. However, despite the presence of both systemic and mucosal-neutralizing antibodies, EAV evades the local host immunity, but the mechanism by which persistence is maintained is unknown [42]. Likewise, some amino acid exchanges affecting viral virulence in cell culture also locate to the ectodomain of GP5 [43].

Despite its critical role in virus replication and immune evasion, the three-dimensional structure of the GP5/M complex remains unknown. Structural insights are essential for understanding protein function, interactions, and disease mechanisms. While traditional methods like X-ray crystallography and Cryo-EM have been invaluable, they are time-consuming and technically demanding, especially for protein complexes with several transmembrane regions. AI-driven tools like AlphaFold2 have transformed protein structure prediction, achieving atomic-level accuracy as demonstrated in the 2020 CASP competition [44]. Trained on all experimentally resolved structures in the Protein Data Bank (PDB), AlphaFold2 uses multiple sequence alignments (MSAs) to identify co-varying residues—likely to interact or be spatially close due to evolutionary constraints. It outputs predicted atomic coordinates in mmCIF or PDB formats, along with confidence metrics such as per-residue pLDDT scores, PAE matrices for component positioning, and overall pTM and ipTM scores [45]. Its successor, AlphaFold3, extends predictions to biomolecular complexes involving proteins, ligands, nucleic acids, and post-translational modifications [46]. AlphaFold2 is accessible via open-source code or platforms like ColabFold, which require minimal computational resources, while AlphaFold3 is available through a web-based server for non-commercial use. Previously, we used ColabFold of AlphaFold2 to model GP5/M of PRRSV with high accuracy, but predictions for EAV

were less reliable [32]. In this study, we employed AlphaFold3 to predict the GP5/M structure of EAV, focusing on known antibody epitopes and virulence factors. Knowing the precise 3D structure of an antigenic protein enables researchers to identify surface-exposed epitopes. This enables the design of vaccines that focus the immune response on the most protective parts of the virus, rather than irrelevant or hidden regions. Structural insights also enable the engineering of stabilized antigen variants that accurately mimic the native viral conformation, enhancing immunogenicity [47,48].

## Results

### Predicted signal peptide cleavage in EAV GP5 reveals a conserved site compared to multiple sites in PRRSV

GP5 from distinct PRRSV strains can be cleaved at different sites, and GP5 from a single strain can be cleaved at two separate sites, according to experimental and bioinformatic data [30–32]. It has been suggested that depending on the site of cleavage, a "decoy epitope" is removed from GP5 or still present in virus particles [49,50]. To determine whether a similar heterogeneity exists for GP5 of EAV we predicted the signal peptide cleavage sites of 210 EAV GP5 sequences using SignalP 5.0. With a high probability score of 98 + −1,2%, almost every GP5 protein is expected to be cleaved between positions 18 and 19. Only one GP5 sequence is predicted to be cleaved between residues 16 and 17 and no second cleavage site was predicted for any GP5 protein.

Whether and where a signal peptide is cleaved by the signal peptidase depends primarily on the presence of small and neutral amino acids at the −3 and −1 position with respect to the cleavage site [51]. The web logo displaying the frequency of amino acids at the N-terminus of 210 GP5 sequences shows a conserved pair of the small amino acids Ala and Ser at positions 16 and 18, which correspond to residues −3 and −1 upstream of the cleavage site. Positions 14 and 16 also contain a conserved pair of small amino acids, Val/Ala and Ser, respectively, but the conserved Pro, a disruptor of secondary structures, at position 15 may render this site unsuitable for cleavage (S1 Fig). Thus, the signal peptide of GP5 of EAV (18 amino acids) is substantially shorter than that of GP5 of PRRSV [31–35], and it does not appear that there are more than one signal peptide cleavage sites within one GP5 protein or different cleavage sites between GP5 from different EAV strains. Note also that there are no putative N-glycosylation sites near the signal peptide cleavage site, that could, if used, prevent signal peptide cleavage, as shown for GP3 of EAV [52,53].

### Quality evaluation of EAV GP5/M model and structural comparison to PRRSV

We employed AlphaFold3 to predict the three-dimensional structure of the GP5/M dimer of the equine arteritis virus (EAV) reference strain Bucyrus, using the GP5 sequence excluding its signal peptide and the complete amino acid sequence of the M protein. AlphaFold3 generated five structure predictions, each accompanied by metrics to assess prediction quality.

The predicted template modeling (pTM) score of the best model, which evaluates the overall structural accuracy, was 0.48, slightly below the 0.5 threshold typically indicative of a reliable global fold. This suggests that the predicted structure of the GP5/M dimer may approximate the true conformation but warrants cautious interpretation. The transmembrane regions of the GP5/M dimer and a portion of the GP5 ectodomain exhibit pLDDT scores between 70 and 90, reflecting high confidence in the main-chain predictions for these regions. Other segments of the GP5 ectodomain exhibit pLDDT scores between 50 and 70, indicating moderate confidence where secondary structural elements, such as α-helices and β-strands, are likely correct, though their spatial arrangement may be less precise. The endodomains show pLDDT scores below 50, suggesting low confidence, potentially due to prediction uncertainty in these regions or intrinsic disorder (S2A Fig).

To visualize residue-level confidence, we also mapped pLDDT scores onto the predicted structure using a rainbow color gradient, with red indicating high confidence (pLDDT > 90) and blue denoting low confidence (pLDDT < 50). This representation highlights that the membrane spanning parts of all six transmembrane regions and a prominent β-sheet and the following helix 3 within the GP5 ectodomain are predicted with the highest confidence, underscoring their structural reliability (Fig 1A, 1B, 2A).

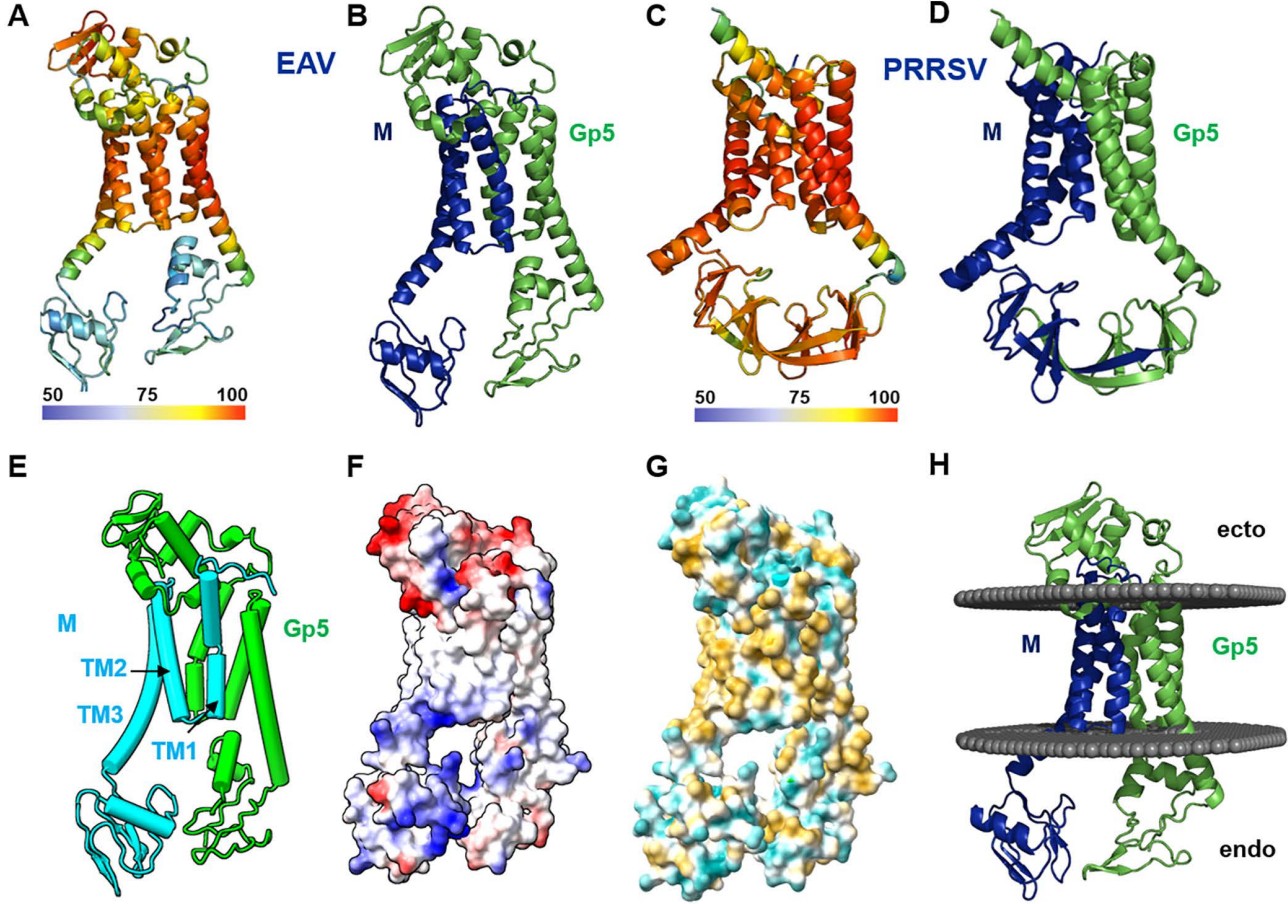

**Fig 1. Structural analysis of the Gp5/M dimer of EAV reference strain Bucyrus. (A**) Per-residue confidence (pLDDT) for the AlphaFold3-predicted Gp5/M dimer of EAV, visualized in a rainbow color gradient from red (high confidence, pLDDT > 90) to blue (low confidence, pLDDT < 50), as indicated by the accompanying color bar. (**B**) Cartoon representation of the EAV Gp5/M dimer, with Gp5 in green and M in blue. (**C**) Per-residue pLDDT scores for the predicted Gp5/M dimer of PRRSV-2 strain VR-2332, displayed in the same rainbow color scheme. (**D**) Cartoon representation of the PRRSV-2 Gp5/M dimer, with Gp5 in green and M in cyan. (**E**) Simplified schematic of the EAV Gp5/M structure, depicting helical regions as cylinders and β-strands as arrows, with Gp5 in green and M in light blue. (**F**) Electrostatic surface potential of the EAV Gp5/M dimer, with acidic regions in red and basic regions in blue. (**G**) Hydrophobic surface representation of the EAV Gp5/M dimer, with hydrophilic residues in light blue and hydrophobic residues in yellow-brown. (**H**) The EAV Gp5/M dimer embedded in a virtual Golgi membrane bilayer, illustrating the positioning of transmembrane helices.

The second confidence metric, called the prediction aligned error (PAE), measures confidence in the relative positions of pairs of residues (S3 Fig). The PAE is displayed as a 2D plot, and the expected position error in Angstrom is color-coded from dark green (0Å) to light green (30Å). Five regions exhibit low PAE values of 3 Å or less, indicating high confidence in their relative positioning: (i) intra-ectodomain distances within GP5, (ii) intra-transmembrane distances within GP5, (iii) intra-transmembrane distances within the M protein, (iv) inter-transmembrane distances between GP5 and M, and (v) inter-transmembrane interactions between M and GP5. Distances between the GP5 ectodomain and the transmembrane regions of both proteins show moderate PAE values, suggesting acceptable but less precise predictions. In contrast, the endodomain regions display high PAE values, ranging from 25 to 30 Å, indicating substantial uncertainty in their relative positioning, likely due to intrinsic disorder or limitations in prediction accuracy. Overall, the PAE analysis confirms that parts of the ectodomain and the transmembrane regions of the GP5/M dimer are predicted with reasonable structural reliability, while the endodomain predictions are of low quality.

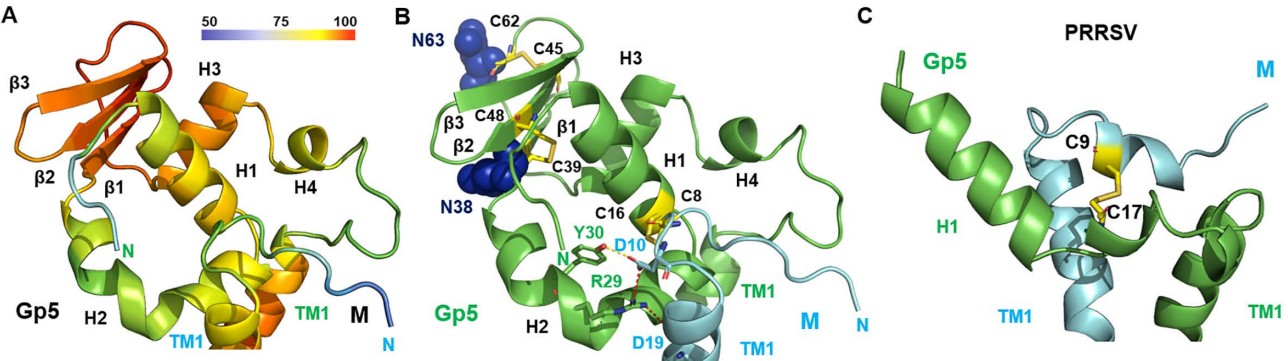

**Fig 2. Structural features of the Gp5/M ectodomain of EAV. (A)** Cartoon representation of the Gp5/M ectodomain of EAV, colored by per-residue pLDDT confidence scores in a rainbow gradient from red (high confidence, pLDDT > 90) to blue (low confidence, pLDDT < 50). Predicted α-helices (H) and β-strands (β) are numbered sequentially from N- to C-terminus; TM denotes the transmembrane region. **(B)** Cartoon model of the ectodomain, with Gp5 in green and M in cyan. Glycosylation sites (N38, N63) are depicted as blue sticks. Cysteine residues involved in predicted disulfide bonds (C39–C48, C45–C62) or experimentally confirmed bonds (C16 in Gp5–C8 in M) are shown as yellow sticks. Additional interactions include a hydrogen bond (Y30 in Gp5–D10 in M) and two salt bridges (R29–D10, R29–D19). **(C)** Cartoon model of the ectodomain of the PRRSV-2 prototype virus VR 2332, with Gp5 in green and M in cyan. Cysteine residues involved in the experimentally confirmed disulfide bond (C17 in Gp5–C9 in M) are shown as yellow sticks.

Generating a Ramachandran plot from the PDB-file revealed that 91,2% of amino acids are in the most favoured regions, 8,3% in the additional allowed region, 0.6% (2 residue) in the generously allowed region and none in the disallowed region (S2B Fig). Generally, good quality models have over 90% in the most favoured region, again confirming the validity of the predicted model [54].

To evaluate structural convergence and ensure the reliability of the predictions, we performed a two-tiered comparison: first, between the models with the highest and lowest confidence scores within the initial run, and second, between the top-ranked models from two independent AlphaFold 3 runs. Unlike traditional deterministic modeling, AlphaFold 3 utilizes a generative diffusion process that constructs structures by iteratively 'denoising' a cloud of initial random atomic coordinates. This stochastic process is initiated by a 'seed", a numerical starting point for the random number generator. While a single prediction run generates five distinct diffusion samples from the same seed, performing independent runs provides an ensemble of predictions from different starting seeds. Consistency across these samples and seeds indicates a robust, well-defined fold, whereas structural variability identifies regions of either intrinsic physical flexibility or lower predictive confidence.

All three models exhibit the same set of structural elements in the same order, with the only notable exception being the M-protein N-terminus, which forms a short α-helix in one model but remains unstructured in others. Mapping of pLDDT scores confirms that the six transmembrane regions, the β-sheet, and helix 3 of the GP5 ectodomain are predicted with high confidence and align closely across all models. A minor misalignment is present in helix 4 across all models. A more pronounced deviation is observed between helices 1 and 2 when comparing the models from the two runs. Despite these localized variations in flexible loops and the N-terminus, the high structural convergence and confidence scores across the transmembrane core and key ectodomain elements demonstrate that the predicted heterodimer is robust (S4 Fig).

We previously predicted the structure of the GP5/M dimer of porcine reproductive and respiratory syndrome virus (PRRSV) [32]. The PRRSV model exhibits notably higher predicted local distance difference test (pLDDT) scores, particularly in the endodomain, than that of EAV GP5/M dimer (Fig 1C, 1D). This difference likely stems from the greater availability of PRRSV sequences in the GenBank database, over 1,000 full-length GP5 sequences and roughly 100 M sequences contributed to the multiple sequence alignment (MSA), providing a richer dataset for identifying covarying amino acid pairs during evolution—a critical factor for accurate structure prediction in AlphaFold [32]. In contrast, the EAV

dataset is both smaller and often incomplete, approximately 100 M and 450 GP5 sequences in total, with only about 250 GP5 sequences covering the central region, constrain the reliability of the EAV model, especially in the endodomain.

Despite these differences, the predicted GP5/M dimer structures of EAV and PRRSV share similarities. Both feature short ectodomains, followed by three transmembrane helical regions, while their endodomains are predominantly composed of β-strands. In the PRRSV model, the endodomains of GP5 and M interact primarily through the longest β-strand (β7), forming an antiparallel β-sheet [32]. No such interactions are predicted in the EAV model, likely due to the low-confidence predictions in the endodomain region.

When using AlphaFold3 instead of AlphaFold2, we observed a substantial improvement in the predicted structure of the EAV GP5/M dimer. In the AlphaFold2 model, only the β-sheet and the transmembrane helices TM2 and TM3 of GP5 were predicted with good confidence, and no interactions between GP5 and M, including the conserved disulfide bond, were recovered. In addition, no meaningful structural alignment between the AlphaFold2 models was possible (S5 Fig). To evaluate whether similar improvements occur for PRRSV, we also recalculated the GP5/M dimer using AlphaFold3. In contrast to the EAV GP5/M model, the overall quality scores were comparable to the AlphaFold2 prediction, although regions with low to moderate confidence were located in different parts of the protein. Structural alignment of the two models yielded an RMSD of ~2 Å, with the only notable difference occurring in the helical N-terminus of GP5 (S6 Fig).

Analysis of the electrostatic surface potential of the EAV GP5/M dimer reveals an acidic ectodomain and a basic endodomain. The central region of the complex, corresponding to the transmembrane domains, lacks significant surface charges and is predominantly hydrophobic, as evidenced by a hydrophobic surface representation. When the predicted structure was oriented within a virtual Golgi membrane bilayer, the three α-helices of each protein were positioned within the hydrophobic core of the lipid bilayer, consistent with their transmembrane nature (Fig 1E-1H).

## GP5/M ectodomain in EAV contains unique structural elements absent in PRRSV

The ectodomain of GP5 in equine arteritis virus (EAV) is predicted with moderate to high confidence, as indicated by reliable pLDDT scores and low predicted aligned error (PAE) values (Fig 2A, S3 Fig). In both EAV and PRRSV, the M protein ectodomain is short, comprising only 10 amino acids, whereas the GP5 ectodomain of EAV is significantly longer (89 amino acids) than that of PRRSV (26 amino acids). The extended GP5 ectodomain of EAV may influence receptor engagement and other host-factor interactions, which aligns with the broader cell and tissue tropism of EAV. Multiple sequence alignments of the EAV GP5 ectodomain with PRRSV-1 (Lelystad) and PRRSV-2 (VR-2332) reveal an insertion in EAV GP5 and few conserved residues only in the N-terminal region (residues 6–20, (S7 Fig)), which forms an upright α-helix in both viruses (Fig 2B, 2C). This helix includes a conserved cysteine (Cys16 in GP5) that forms an experimentally validated disulfide-bond with Cys8 in M [32]. This disulfide linkage is correctly predicted in the EAV-model, further confirming its accuracy. Note that our numbering corresponds to the mature GP5 protein without signal peptide.

The extended length of the EAV GP5 ectodomain introduces additional structural features absent in PRRSV, including three α-helices and a three-stranded β-sheet inserted between helix 2 and helix 3. Helix 2, which is kinked and extends upward, connects to the highly confident three-stranded β-sheet at the molecule's apex, stabilized by two disulfide bonds: one linking β1 to β2 (Cys39–Cys48) and another connecting β3 to a loop (Cys45–Cys62). In the Bucyrus strain of EAV, two N-glycosylation sites are present: Asn38 on β-strand 1 and Asn63 in the loop region connecting the β-sheet to helix 3. Notably, some laboratory-adapted EAV strains lack Asn63, suggesting it is non-essential for GP5 function [43]. The protein chain then folds back toward the membrane via helix 3 and a short helix 4, linking to the transmembrane region (Fig 2B).

## Mapping of antibody epitopes in GP5's ectodomain and their strain variability: Implications for immune evasion via antigenic drift and glycan shielding

Initial analyses of viral mutants that escape neutralisation by monoclonal antibodies identified four discrete epitopes within the GP5 ectodomain [36,37]. Two of these, designated sites A and B, correspond to epitopes centred on amino acids 31

and 43, respectively. The remaining sites, C and D, are broader regions spanning residues 49–72 and 80–88, respectively. Notably, site C, the longest epitope, encompasses the β2 and β3 strands, the loop connecting to helix 3, and the initial residues of helix 3 (Fig 3A). A semi-transparent surface projection of the GP5/M dimer reveals that most epitope residues are surface-exposed, as anticipated. Furthermore, sites B and C are positioned in close proximity at the molecule's apex, suggesting they form a continuous conformational epitope. In contrast, sites A and D, located in helix 1 and helix 4, respectively, are spatially separated in the three-dimensional model (Fig 3B, 3C). Note that the β-sheet and the following helix 3 are predicted in all models with the highest accuracy and their structures align well (S4 Fig).

Subsequent analyses using recombinant EAV mutants with targeted GP5 substitutions confirmed both linear and conformational epitopes at residues 31 (site A), 43 (site B), 51, 81, 84, 85, and 86 [35] Immunization of horses with peptides corresponding to amino acids 75–97 elicited neutralizing antibodies, indicating surface exposure of this epitope on the intact virion [38]. The latter four residues, situated in or near helix 4, fall within site D. More recent peptide library screening revealed that the region spanning residues 57–71 exhibited the highest reactivity with sera from infected horses, with 55% of samples showing positive responses [39] This region encompasses the loop connecting the β-sheet to helix 3, closely aligning with site B (Fig 3D–3F).

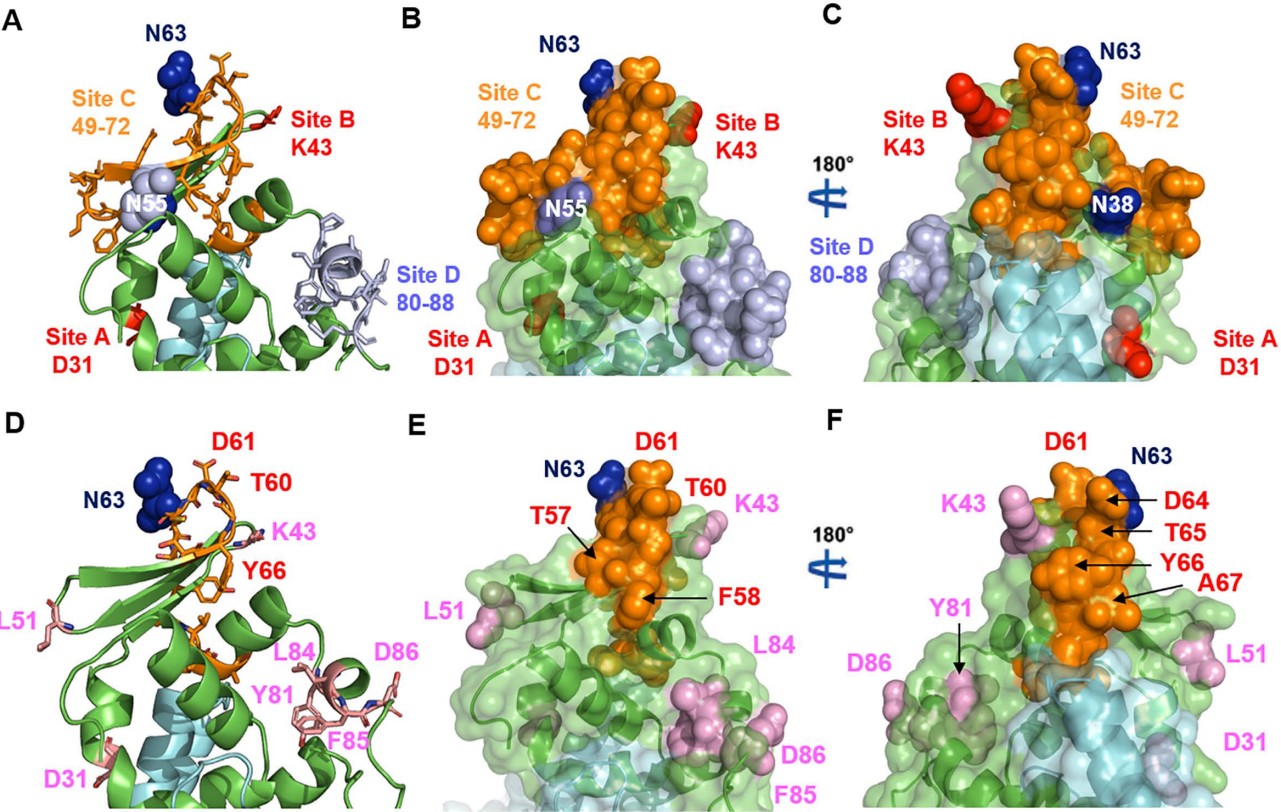

**Fig 3. Antibody epitopes of the Gp5/M ectodomain of EAV. (A-C)** Cartoon representation (A) and semi-transparent surface representation of the ectodomain **(B,C)**, highlighting antibody epitopes as sticks (A) or as spheres **(B,C)**: site A and site B in red, site C in wheat, and site D in orange. Glycosylation sites (N38, N63) are shown as cyan spheres. Panel C is rotated 180° relative to panel **B. (D-F)** Cartoon representation (D) and semi-transparent surface representation of the ectodomain **(E, F)**, highlighting other antibody epitopes as sticks (D) or as spheres **(E,F)**: site A and site B in red, site C in wheat, and site D in orange. Glycosylation sites (N38, N63) are shown as cyan spheres. Panel C is rotated 180° relative to panel **B**. Note: Amino acid numbering reflects the mature protein, excluding the 18-amino-acid signal peptide; add 18 to calculate positions in the full Gp5 sequence.

Neutralizing antibody epitopes often exhibit amino acid sequence variability across strains due to antigenic drift. To assess this variability among equine arteritis virus (EAV) strains, we employed two analytical tools: (i) ConSurf, which estimates and visualizes evolutionary conservation in macromolecules using 250 M and 208 GP5 sequences retrieved from UniRef90, and (ii) WebLogo, which quantifies amino acid frequency at each position. Variant residues determined with ConSurf are shown as sticks, coloured form deep to light blue according to their variability, in the cartoon projection of the GP5 ectodomain (Fig 4A). The highest variability was observed in the β-sheet at the molecule's apex, particularly in the loops connecting individual strands and linking the β-sheet to helix 3. Structurally, this is expected, as loops tolerate greater sequence variation without disrupting local secondary structure. This region corresponds to epitope site C. Similarly, residue 42 and its flanking residues, corresponding to site B, exhibit significant variability. Helix 4, encompassing site D, also contains multiple variable residues. A semi-transparent surface projection indicates that many variable residues are surface-exposed, suggesting positive selection driven by antibody binding (Fig 4B). In contrast, residue 31 (site A) is invariant and less surface-accessible, casting doubt on its role as a neutralizing epitope. Notably, the ectodomain of the M protein consists exclusively of invariant residues, indicating it likely does not contribute to immune evasion (S8A Fig).

The highly variable epitopes B and C are situated in close proximity to N-glycosylation sites Asn38 and Asn63 (depicted as spheres in Fig 3). The sequon $_{38}NCS_{40}$ is highly conserved, whereas the sequon $_{63}NDT_{65}$ exhibits modest variability including loss of the N-glycosylation site (Fig 4C). These carbohydrates, particularly at Asn63, may partially restrict antibody access to these epitopes. Additionally, an N-glycosylation site at Asn55, located in β-strand 2 at the molecule's apex, was identified in isolates from a major EAV outbreak in North America [55] (depicted as sphere in Fig 3). Residue 55 is variable, typically occupied by isoleucine or valine, which represent a conservative substitution, and is followed by invariant residues Ile56 and Thr57 (Fig 4C). The rare substitution of asparagine at position 55 forms the N-glycosylation motif $_{55}NIT_{57}$, which, if glycosylated, could shield portions of epitope C. These findings suggest that glycan modifications may contribute to immune evasion by masking critical epitopes. Collectively, the three-dimensional model of the GP5/M dimer indicates that antigenic drift, combined with glycan shielding, likely represents a key mechanism by which EAV evades antibody recognition.

## Mapping virulence and persistence determinants in the GP5 ectodomain

The predicted structure of the GP5 ectodomain was used to map molecular determinants of viral virulence and persistence (Fig 5A, 5B). During genomic cloning of the Bucyrus strain isolate, an unintended substitution of Asp86 to Asn in helix 4 resulted in viral attenuation [43]. In a cell culture-passaged attenuated variant of the same strain, Asp86 is substituted with glycine, accompanied by two additional substitutions in GP5: Ser82 to Gly in helix 4 and Asn63 to Asp, which abolishes the N-glycosylation site at the molecule's apex [43]. Furthermore, an equine arteritis virus (EAV) variant capable of establishing persistent infection in HeLa cells harbors a Pro80 to Leu substitution [56].

Most determinants of virulence and persistence are localized within or near helix 4, a short helix connected to helix 3 by a loop containing glycine and proline residues (sequence $_{69}$PVAEVLEQAHG**P**YS**S**ALF**D**DMPP$_{90}$ with helix 3 and 4 underlined and the mutated residues in bold). The observed substitutions either introduce glycine (Ser82Gly, Asp86Gly) or eliminate proline (Pro80Leu), residues known to influence secondary structure. Since Alphafold should not be used to assess the effect of single amino acids exchanges on the structure of an entire protein [57] we predicted only the region encompassing helices 3 and 4 using wild-type and mutant sequences. All models exhibited high confidence, as indicated by high per-residue local distance difference test (pLDDT) scores, visualized in red on the predicted structures. The wild-type sequence predicts two distinct helices with boundaries consistent with the full protein model but with a distinct inter-helix angle. Substitutions Ser82Gly and Asp86Gly also predict two helices with an altered inter-helix angle. However, the Pro80Leu substitution results in the fusion of helices 3 and 4 into a single, elongated helix (Fig 5C–5E). Note that the spatial position of helix 4 varies among the AlphaFold models, but it consistently remains connected to helix 3 by a loop, and helix 3 itself aligns very well across all models (S4 Fig.).

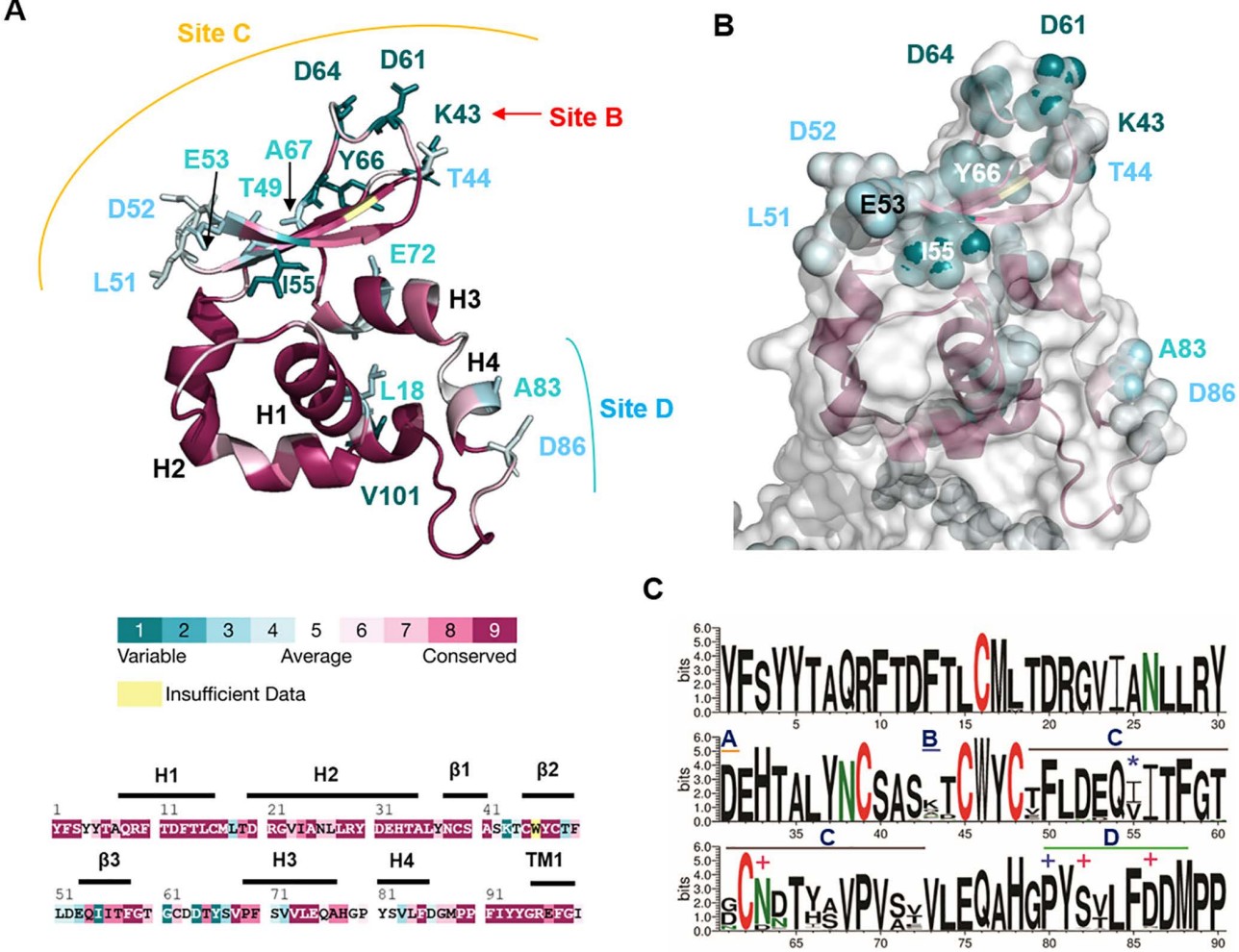

**Fig 4. Conservation of amino acids in the ectodomain of Gp5/M. (A, B)** Three-dimensional structure of the Gp5 and M ectodomains shown as a cartoon representation (A) and a semitransparent surface projection **(B)**. Residues are color-coded based on evolutionary conservation scores calculated using ConSurf. Variable residues are depicted as sticks (A) or spheres (B) and labeled. The color scale ranges from variable (turquoise) to highly conserved (maroon); residues lacking sufficient data are shown in yellow. Conservation scores are also mapped onto the amino acid sequence of the Gp5 ectodomain. **(C)** Web logo showing the amino acids at each position in the ectodomain of Gp5. Logos were generated from 210 aligned amino acids sequences. The overall height of the stack indicates the sequence conservation at a position (X-axis), while the height of symbols within the stack indicates the relative frequency of each amino. Asn residues are labelled green, Cys in red. The epitopes are marked with a line, site A in orange, site B in blue, site C in black and site D in green, with each line labelled accordingly. The amino acids involved in pathogenicity and persistence are labelled with +. The location of an N-glycosylation site found in isolates of an extensive EVA outbreak in North America with *.

## GP5/M transmembrane region in EAV shares structural features with PRRSV and SARS-CoV-2 M and Orf3a

Positioning of the GP5/M dimer of equine arteritis virus (EAV) within a virtual lipid bilayer reveals that the first transmembrane helix (TM1) of both GP5 and M enters the membrane at an approximately 20-degree tilt relative to the membrane normal (Fig 6A). A proline residue (Pro25 in M, Pro110 in GP5, depicted as magenta sticks) induces a kink in the middle of each TM1 helix, causing the lower half of the helices to align nearly parallel to each other and perpendicular to the membrane plane. Short loops (3–4 amino acids) within the bilayer connect TM1 to TM2. Unlike TM1, TM2 is unbent but also adopts a tilted orientation through the membrane. A partially exposed loop links TM2 to TM3, the longest helix, which

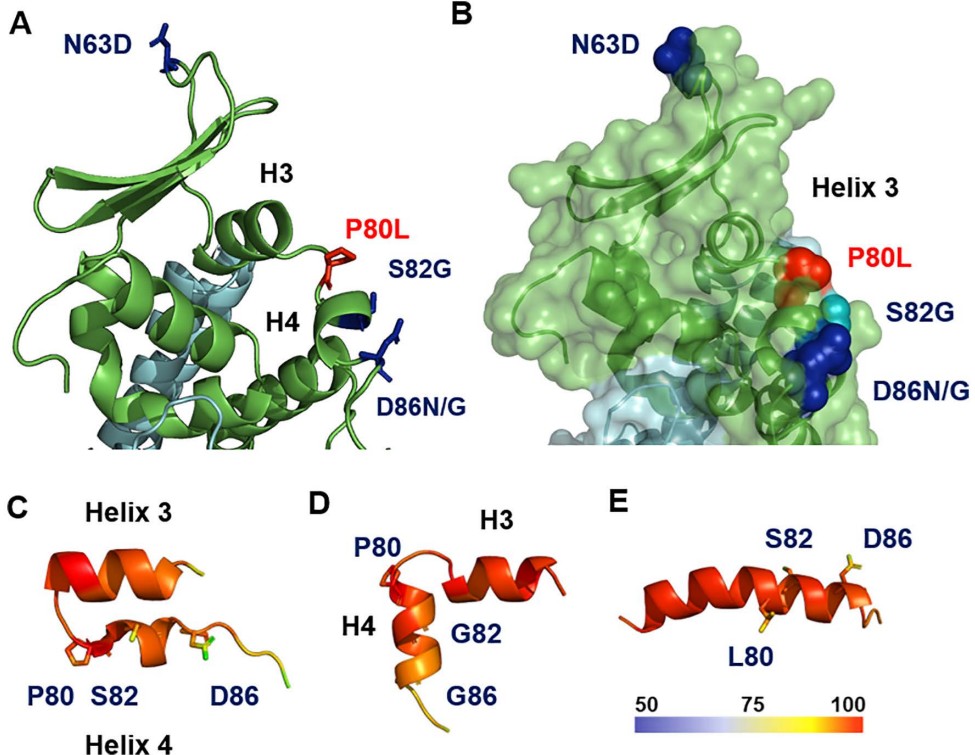

**Fig 5. Amino acid residues in the Gp5/M ectodomain of EAV associated with virulence and persistence. (A–B)** Structure of the Gp5/M ectodomain shown as a cartoon model (A) and a semi-transparent surface projection **(B)**. Residues implicated in pathogenicity and persistence are highlighted as sticks (A) or spheres **(B)**. Blue indicates amino acid substitutions distinguishing the virulent Bucyrus strain from attenuated variants derived through cell culture passage. Red marks substitutions associated with a variant capable of establishing persistent infection in cell culture. **(C–E)** Predicted structures of the region encompassing helices 3 and 4, generated using AlphaFold3. (C) Wild-type sequence. (D) Sequence containing S82G and D86G substitutions found in attenuated, cell-adapted virus. (E) Sequence with the P82L substitution identified in a persistent virus variant. Cartoon models are color-coded by predicted local distance difference test (pLDDT) scores, displayed in rainbow gradient to indicate confidence per residue position.

traverses the membrane at an angle and extends into the cytoplasm. In GP5, but not M, two cysteine residues at the C-terminal end of TM3 serve as potential acylation sites, consistent with fatty acid modifications observed in the GP5/M dimer of PRRSV [34].

Notably, TM2 of GP5 contains a charged residue, Glu128, at its midpoint, which is unusual since charged residues are rarely preset in transmembrane regions. This residue is stabilized by two ionic bonds with Arg32 in TM1 of M, neutralizing its charge and enabling its integration into the hydrophobic bilayer. Additional electrostatic interactions stabilize the GP5/M interface: Asp104 in the outer leaflet of TM1 of GP5 forms a hydrogen bond with Tyr54 in TM2 of M, and Ser118 in the inner leaflet of TM1 of GP5 forms a hydrogen bond with Arg82 in TM3 of M. A ConSurf and WebLogo analysis revealed that all critical residues, including those involved in these interactions, are conserved in GP5 and M of EAV (S8, S9 Fig). Interestingly, all three models predict identical electrostatic interactions between the transmembrane segments of GP5 and M, indicating that the prediction of the interface is robust (S10 Fig).Structural alignment of the transmembrane regions of EAV and PRRSV-2 GP5/M dimers yields a root-mean-square deviation (RMSD) of 3.43 Å, indicating high structural conservation (Fig 6B, 6C). Most transmembrane helices in EAV align closely with their PRRSV counterparts, except for the outer leaflet portions of TM1 and TM2 of M, which diverge slightly. Conserved features include charged residues at equivalent positions in PRRSV, such as an ionic bond between Asp58 in TM2 of GP5 and Lys35 in TM1 of M,

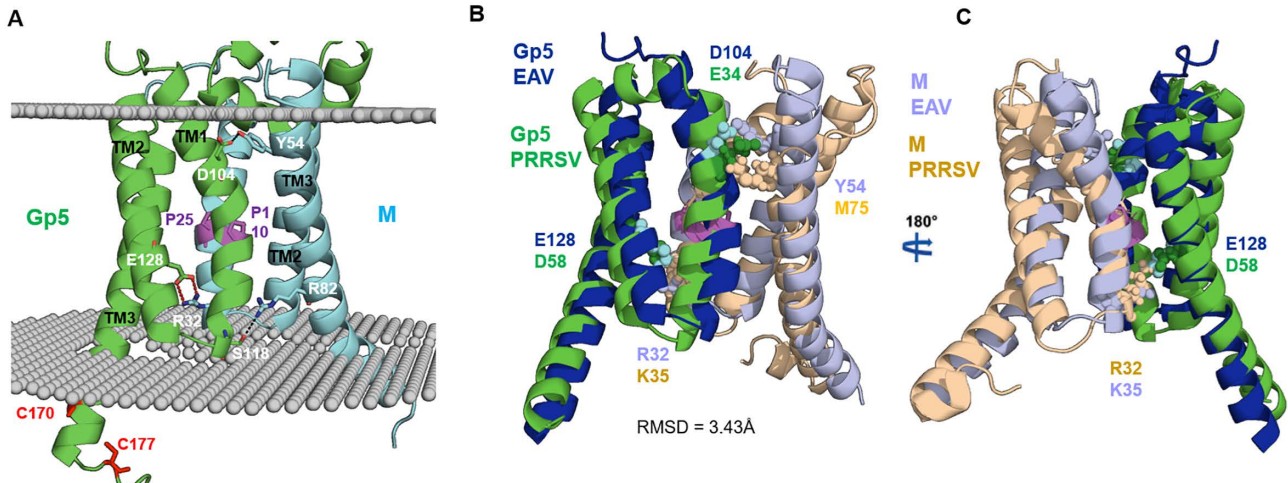

**Fig 6. Structural features of the transmembrane region of Gp5/M from EAV. (A)** Side view of the Gp5/M transmembrane region embedded in a virtual lipid bilayer. Key residues are shown as stick models and labeled. Pro25 in M and Pro110 in Gp5 introduce kinks in transmembrane helix 1 (TM1) of their respective proteins. Glu128 in TM2 of Gp5 forms ionic interactions with Arg32 in TM1 of **M.** Tyr54 and Arg82 in TM3 of M participate in hydrogen bonding with Asp104 and Ser118 in TM1 of Gp5, respectively. Cys170 and Cys177 are proposed sites of acylation. **(B, C)** Structural alignment of the transmembrane domains of Gp5/M from EAV and PRRSV-2. Residues involved in hydrophilic interactions within the lipid bilayer are shown as spheres and labeled. Proline residues in TM1 are highlighted as magenta sticks. RMSD values (in Å) indicate the degree of structural similarity between aligned domains.

and a hydrogen bond between Glu34 in TM1 of GP5 and Met75 in TM3 of M (unlike EAV, where the equivalent interaction involves TM2). Additionally, conserved proline residues in TM1 of both GP5 and M induce a characteristic kink, and acylation sites in GP5 (but not M) are preserved across both viruses. Together, these conserved motifs are consistent with stabilizing interactions within the GP5/M transmembrane region of arteriviruses.

The predicted structure of the GP5/M heterodimer exhibits notable topological and structural similarities to the homodimeric membrane proteins M and Orf3a of SARS-CoV-2, both of which play key roles in viral assembly and budding. These proteins share a conserved architecture comprising a short ectodomain, three transmembrane helices per subunit, and a β-sheet-rich endodomain. Additional shared features include hydrophilic inter-subunit interactions within the membrane, acylated cysteine residues in Orf3a, and a kink-inducing proline residue within one of the transmembrane helices of M. Attempts to align the GP5/M transmembrane regions with those of the long or short M isoforms did not produce meaningful structural matches, while alignment with ORF3a resulted in an RMSD of approximately 8 Å. This mismatch is attributable to distinct interhelical angles that alter the overall topology of the transmembrane bundle (S11 Fig).

To further explore these relationships, we conducted both single and multiple sequence alignments comparing GP5 and M from EAV with their counterparts in the PRRSV-1 prototype strain Lelystad, PRRSV-2 prototype strain VR-2332, and the M and Orf3a proteins of SARS-CoV-2 (S12 Fig). Among these, EAV M showed the highest sequence conservation with PRRSV-1 M (26% identity, 40% similarity) and PRRSV-2 M (24% identity, 37% similarity). Notably, substantial sequence similarity was also observed with SARS-CoV-2 M (18% identity, 28% similarity). Although slightly lower, meaningful sequence similarity was detected between EAV M and GP5 proteins from arteriviruses (14% identity, 24% similarity for EAV; 22%/31% and 16%/26% for PRRSV-2), as well as with SARS-CoV-2 Orf3a (12% identity, 24% similarity). GP5 from EAV displayed comparable homology with GP5 from PRRSV-1 (17% identity, 26% similarity) and PRRSV-2 (17% identity, 30% similarity), and also with PRRSV M proteins (21%/30% for PRRSV-1 and 17%/27% for PRRSV-2) and with SARS-CoV-2 Orf3a (12/19). These findings were consistent across both single and multiple sequence alignment analyses.

Amino acid sequence similarities of approximately 30% or higher are generally indicative of conserved protein folding. Similarities in the 20–30% range fall within a transitional zone where structural conservation is plausible but not guaranteed. Taken together, the relatively high sequence conservation and conserved structural features among the GP5/M heterodimer, M homodimer, and Orf3a homodimer suggest a deep evolutionary relationship across members of the *Nidovirales* order.

## Discussion

Using AlphaFold3, we generated a structural model of the GP5/M dimer from the equine arteritis virus (EAV) Bucyrus strain, revealing similarities to the previously predicted structure of the porcine reproductive and respiratory syndrome virus (PRRSV) GP5/M complex (Fig 1) [32]. Both consist of short ectodomains, three transmembrane helices, and β-sheet-rich endodomains. The interaction between GP5 and M is mediated by an experimentally verified disulfide bond within the ectodomain, as well as contacts within the transmembrane regions. Unlike the PRRSV model, the EAV structure showed no predicted endodomain interactions, likely due to low-confidence modeling in this region—leaving open the possibility of such interactions in the native complex (S2, S3 Fig). The EAV GP5 ectodomain is notably extended, featuring four additional α-helices and between them a disulfide-stabilized β-sheet at its apex, which is predicted with the highest accuracy (Fig 2). The existence of this additional domain in GP5 of EAV suggests that it may confer additional functions to the virus compared to PRRSV. Interestingly, residues 34–61, encompassing the entire β-sheet plus the first part of the loop, can be removed from GP5, and the resulting recombinant virus is still capable of growing in cell cultures, albeit with a significant reduction in titer [33]. This implies that while this domain is not essential for viral replication in cell culture, it likely plays a role in optimizing viral growth and is important for virus replication *in vivo*.

Another notable distinction is the length of the GP5 signal peptide, which is markedly shorter in EAV than in PRRSV (18 versus 31–35 amino acids). Our bioinformatic analysis of 210 EAV strains consistently identified a cleavage site between residues 18 and 19, suggesting a highly uniform processing mechanism (S1 Fig). In contrast, PRRSV exhibits substantial variability in GP5 signal peptide cleavage, a feature that may influence the presence or absence of a decoy epitope on viral particles [31,49,50].

Neutralizing antibody epitopes previously identified through peptide-based assays were mapped in this study onto the predicted 3D structure of the GP5 ectodomain. Site C—overlapping with regions in recombinant EAV mutants that escape antibody recognition—and the single-residue site B both exhibit high surface accessibility and are located in close spatial proximity at the apex of the molecule, suggesting they may form a continuous conformational epitope (Fig 3). Sequence variability analysis revealed pronounced diversity within the β-sheet region and its connecting loops corresponding to site C, as well as in site B, which consists of a single, highly variable surface-exposed residue (Fig 4). This elevated variability is consistent with positive selection driven by antibody-mediated immune pressure. Furthermore, the proximity of epitopes B and C to the conserved N-glycosylation site Asn38, the moderately variable site Asn63, and an outbreak-associated glycosylation site at Asn55 supports a mechanism of immune evasion involving antigenic drift in combination with glycan shielding.

Given the high surface accessibility and spatial proximity of epitopes B and C at the apex of the GP5 ectodomain, recombinant proteins representing this region may serve as promising vaccine candidates. Although these epitopes are characterized by high sequence variability, our model indicates that they are supported by the β-sheet structure stabilized by disulfide linkages. This structural stability might preserve the native conformation of the epitope and enhance the likelihood of eliciting neutralizing antibodies.. To broaden protective efficacy, the recombinant protein should be engineered to represent the consensus sequence across EAV lineages or incorporate a polyvalent mixture of amino acids at each position that reflect the diversity of epitopes found in currently circulating EAV strains. This design is expected to induce a cross-reactive antibody response capable of neutralizing prevalent viral variants. Moreover, fusing this region to a dendritic cell-targeting peptide could facilitate more robust antibody responses. For practical application, such a vaccine could

be administered via intramuscular injection using modern adjuvants or through viral vector delivery to ensure both robust mucosal protection and systemic immunity.

Epitope site D and the single amino acid site A—both overlapping with peptides known to elicit antibody responses in horses—are positioned near the membrane but on opposite sides of the molecule. Site D, encompassing helix 4, is surface-exposed yet exhibits lower variability compared to the more immunogenic epitopes located at the molecular apex. In contrast, site A, corresponding to residue 31, is invariant and structurally less accessible. The genomic regions encoding GP5 and the ORF5a protein overlap, placing helix 1 and part of helix 2 within this shared reading frame. Specifically, the codon for residue 31 in GP5 coincides with the ORF5a stop codon. Consequently, most nucleotide substitutions at this position would result in a read-through of the ORF5a stop codon, leading to an elongated and likely non-functional ORF5a protein [58,59]. Given that ORF5a is beneficial for EAV infectivity, this dual-coding requirement imposes a severe evolutionary constraint that likely accounts for the observed invariance of site A [58]. These invariant regions, alongside the highly conserved M protein, may therefore represent potential targets for broadly neutralizing antibodies, offering a pathway toward cross-strain protection that is less susceptible to the rapid antigenic drift observed at the molecular apex.

Most amino acid substitutions associated with reduced virulence and persistence in cell culture are localized within the neutralizing epitope D (Fig 5). The relative conservation of this short epitope may reflect functional constraints, as not all residues can undergo mutation without compromising essential structural or functional aspects of the GP5/M dimer. Notably, all characterized viral variants also harbour substitutions in other envelope proteins—particularly within the ectodomains of GP2, GP3, and/or GP4—suggesting that a coordinated interaction between minor and major membrane glycoproteins contributes to the cell tropism of equine arteritis virus (EAV), especially for monocytes and T-cells [60]. This observation supports the hypothesis that the region surrounding helix 4 in GP5 may serve as a docking interface for the minor glycoproteins GP2, GP3, and GP4 [27]. Such a multimeric complex of Arterivirus membrane proteins could facilitate receptor engagement or initiate membrane fusion, thereby promoting viral entry into host cells [61].

The transmembrane domains (TMDs) of the GP5/M heterodimer from EAV exhibit close structural homology to those of PRRSV, with a root-mean-square deviation (RMSD) of 3.43 Å. This conserved architecture is defined by stabilizing hydrophilic interactions, kinked TM1 helices, and acylation sites within GP5 (Fig 6, S5, S6 Fig). In addition to this close relationship, GP5/M shares notable sequence and structural similarities with the homodimeric membrane proteins M and ORF3a from SARS-CoV-2 (S10, S11 Fig).

The SARS-CoV-2 M protein, a key driver of coronavirus budding, adopts two distinct conformational states: a compact "short-form" with a mushroom-shaped topology, and an extended "long-form" configuration [62]. The short-form binds ceramide-1-phosphate (C1-P), a sphingolipid that stabilizes this conformation and facilitates nucleocapsid recruitment to the budding site [63]. The long-form promotes oligomerization through an elongated dimer interface, thereby enhancing membrane curvature and driving virion assembly. Notably, two small-molecule inhibitors targeting the C1-P binding site have been developed; by stabilizing the short-form, these compounds effectively disrupt M protein oligomerization and inhibit viral assembly [64,65].

Emerging evidence also implicates ORF3a in the budding process. This accessory protein orchestrates the formation of "3a dense bodies"—specialized membrane compartments derived from the trans-Golgi network (TGN) and early endosomes. These structures serve as critical assembly platforms for viral structural proteins [66,67].

In summary, this study demonstrates that the AlphaFold3-predicted structure of the EAV GP5/M dimer reliably captures the architecture of parts of the ectodomain and transmembrane regions, offering new molecular insights into its organization. The model enables mapping of virulence-associated determinants and known antibody epitopes, which is consistent with a dual immune evasion strategy that combines antigenic drift with glycan shielding. At the same time, AlphaFold3 predictions reflect conserved topology and overall fold but cannot establish functional or evolutionary relationships. Because the models are computational and static, and several regions, particularly the cytoplasmic endodomains, are predicted with low confidence, functional or mechanistic equivalence cannot be inferred. Within these limitations, the conserved

architecture of the transmembrane domains of Arterivirus GP5/M and those of SARS-CoV-2 M and ORF3a is consistent with, but does not demonstrate, the possibility of a distant shared ancestry or common principles of membrane-associated assembly within the Nidovirales.

## Materials and methods

### Predictions of protein structures with AlphaFold3

The structural prediction of the GP5/M heterodimers from EAV strain Bucyrus (DQ846750.1), and the PRRSV-2 prototype strain VR2332 (AY150564.1) was performed using the AlphaFold 3 algorithm via the AlphaFold Server (https://alphafoldserver.com/; accessed between July 7 and August 17, 2025, and on January 25, 2026). The complete amino acid sequences of the M protein were used as input, whereas for GP5 the predicted N-terminal signal peptides (residues 1–18 for EAV and 1–31 for PRRSV) were removed to improve the accuracy of the ectodomain folding and dimer interface prediction.

Predictions were executed using default server parameters and the 'auto-seed' function. AlphaFold 3 automatically generated its own Multiple Sequence Alignments (MSAs) by searching built-in databases to identify evolutionary constraints and co-varying residues. To evaluate structural convergence, we performed two independent prediction runs for GP5/M of EAV, each generating five stochastic diffusion samples.

Models were automatically ranked by the server based on a composite ranking score, which for multimeric complexes is a weighted combination of the predicted Template Modeling (pTM) score and the interface pTM (ipTM). This score was used to identify the 'best' (highest-ranked) and 'worst' (lowest-ranked) models within each ensemble.

Our validation followed a two-tiered comparison: (i) an intra-run analysis comparing the best and worst models to assess sampling diversity, and (ii) an inter-run analysis comparing the top-ranked models from both independent seeds using Root Mean Square Deviation (RMSD) of Cα atoms.

For prediction with Alphafold2 of Gp5/M of PPRRSV-2 VR 2332 and EAV Bucyrus we used the alphafold2advanced.ipynb notebook, (accessed on 24 and 25 October 2021) [68]. https://colab.research.google.com/github/sokrypton/ColabFold/blob/main/beta/AlphaFold2_advanced.ipynb with the following settings: msa_method = mmseqs2, homoo-ligomer = 1:1, pair_mode = unpaired, max_msa = 512:1024, subsample_msa = True, num_relax = 1, use_turbo = True, use_ptm = True, rank_by = pLDDT, num_models = 5, num_samples = 1, num_ensemble = 1, max_recycles = 3, tol = 0, is_training = False, use_templates = False. The AlphaFold2 predictions were generated using exactly the same amino-acid sequences as those employed for the AlphaFold3 predictions.

The comprehensive data provided in the downloadable output files—including atomic coordinates (PDB/CIF), per-residue confidence scores (pLDDT), Predicted Aligned Error (PAE) matrices, and the generated MSAs—were utilized for all subsequent structural and evolutionary analyses.

To analyse various parameters of the predicted GP5/M structure, such as the Ramachandran plot, ionic bonds, and inter-subunit contacts we used PDB-sum, http://www.ebi.ac.uk/thornton-srv/databases/pdbsum/Generate.html [69].

We used mainly PyMol version 2.1.1 to visualize and analyze the PDB-files made available by AlphaFold. Some figures were created with ChimeraX 1.3.

### Assessment of the quality of the model of the GP5/M dimer

AlphaFold3 generates five structural models and several confidence scores. One, a per-residue confidence metric called the predicted local distance difference test (pLDDT) indicates confidence in the local structure prediction. The scale ranges from 0 to 100 and an IDDT value above 90 indicates very high accuracy, equivalent to structures determined by experiments, which allows for investigating details of individual side chains. A value from 70 to 90 indicates high accuracy, where the predictions of the protein's backbone are reliable. A value of 50–70 indicates lower accuracy, but probably the predictions of the individual secondary structural elements, α-helices, and β-strands

are correct, but how they are aligned in space is uncertain. Values below 50 might be an indication of an intrinsically unstructured region. The PDB-files with the predicted structure contain this information in the B-factors, which can be highlighted in the 3D protein structure. Areas with high B-factors, which indicates high confidence, are colored red, while low B-factors are colored blue.

The second confidence metric, called the prediction aligned error (PAE) measures confidence in the relative positions of pairs of residues. PAE is displayed as a 2D plot and the expected position error in Angstrom is color-coded. The expected position error is usually low for two amino acids present in one domain, but sometimes high for residues located in two domains. This indicates that AlphaFold is uncertain about the relative position of two domains that have no predicted contact with each other, for example, if they are connected by a flexible linker.

The predicted template modeling (pTM) score measures the accuracy of the global structure of the protein and is relatively insensitive to localised inaccuracies. The inter-chain predicted TM-score (ipTM) specifically measures the confidence in the inter-chain interface of a protein complex. pTM and ipTM scores between 0.5 and 0.7 indicate moderate confidence; while the interface might be correct, there could be significant structural deviations. Note that iPTM is only calculated by AlphaFold3 and pTM scores are generated using fundamentally different architectures and hence should be interpreted with caution when making direct quantitative comparisons across versions.

## Calculation of the position of GP5/M within a virtual lipid bilayer

To orient the GP5/M structure in a virtual lipid bilayer, we used the PPM 3.0 Web Server https://opm.phar.umich.edu/ppm_server3 [70].The location of a protein in the membrane coordinate system is obtained by optimization of protein transfer energy (ΔG transfer) from water to a lipid bilayer. Simplified, as many hydrophobic amino acids as possible should be inside the hydrophobic part of the bilayer to avoid hydrophobic mismatch. The anisotropic properties of the lipid bilayer composed of 1,2-dioleoyl-sn-glycero-3-phosphocholine (DOPC) are described by transbilayer profiles of dielectric constant and hydrogen bonding acidity and basicity parameters. The type of membrane was set to a flat Golgi-membrane since this is the main localization of the GP5/M inside cells. The Golgi membrane has a hydrophobic thickness of 30.2 ± 1.3 Å, which roughly corresponds to the length of two acyl chains. Note that the whole bilayer is thicker (~10 Å) since the hydrophilic head groups of the lipids are outside the hydrophobic core domain. As input for the program, the PDB-files of the GP5/M dimer of EAV was used. The program provides a PDB-file of the dimer positioned in the membrane, with the boundaries of the hydrophobic domain marked by dummy atoms, shown as gray spheres. A ΔG transfer of −61.6 kcal/mol was calculated for both structures which lies within the range between −400 and −10 kcal/mol usually calculated for integral membrane proteins.

## Prediction of signal peptide cleavage sites in GP5 of EAV

We used SignalP 5.0, a deep neural network-based approach, to predict signal peptide cleavage sites from GP5 [71]. Supplied with the full-length protein sequence, SignalP provided the information on whether or not the N-terminus of GP5 acts as a signal peptide and also the position at which it is cleaved. The resulting summary sheets list the Uniprot ID for each queried GP5 protein, the prediction of whether the N-terminus is a eukaryotic signal peptide, the prediction of the cleavage site, the five amino acids surrounding the site and the probabilities for each prediction between 0 and 1. The mean including standard deviation for the probabilities and the highest and lowest probabilities for each cleavage site were also calculated. For each predicted sequence a graphical representation is also delivered, which gives the probability for each of the ~70 N-terminal amino acids whether or not it is part of a eukaryotic signal peptide and also whether it functions as a cleavage site. In this way, additional putative cleavage sites can be identified with a lower probability than the main cleavage site. For the prediction, we used the 210 GP5 sequences from EAV gained from the protein blast search described above.

## Sequence conservation analysis

To analyse variation between GP5 sequences from EAV we performed a protein blast search with the GP5 sequence from the EAV reference strain Bucyrus against the non-redundant GenBank database CDS translations+PDB+Swis-sProt+PIR+PRF excluding environmental samples with quick blast. Each of the retrieved 219 sequences was annotated with EAV. Nine incomplete sequences were deleted from the list and the remaining 210 sequences were aligned using Clustal Omega (https://www.ebi.ac.uk/Tools/msa/clustalo/).

To determine and visualize which amino acids are present at a certain position of the sequence and at which frequency, we used web logo http://weblogo.threeplusone.com/. The web-based application generates a graphical representation from a multiple sequence alignment. Each logo consists of stacks of amino acid symbols, one stack for each position in the sequence. The overall height of the stack indicates the sequence conservation at that position, while the height of symbols within the stack indicates the relative frequency of each amino at that position [72]. To keep the numbering of positions in web logo identical to the consensus sequences, we removed the few sequences with insertions from the multiple sequence alignment before it was submitted to the web logo analysis.

## Consurf variability analysis

Evolutionary conservation of amino acid residues in the GP5 ectodomain was analyzed using the ConSurf web server (https://consurf.tau.ac.il, accessed on 25th July 2025) and default parameters [73]. The tool extracted the protein sequence from the PDB-file with the predicted GP5/M structure and 208 GP5 and 250 M homologous sequences were retrieved from the UniRef90 database. A multiple sequence alignment was generated with MAFFT, and a phylogenetic tree was constructed using the neighbor-joining algorithm in Rate4Site. Conservation scores were calculated with the empirical Bayesian method, normalized, and divided into nine grades (1: most variable, turquoise; 9: most conserved, maroon). These scores were projected onto the GP5 three-dimensional structure to identify conserved and variable regions, with results visualized using PyMOL.

## Supporting information

**S1 Fig. Web loge representation of the first 25 amino acids of the full-length Gp5 of EAV strains.**
(PDF)

**S2 Fig. Quality scores for the Gp5/M model.**
(PDF)

**S3 Fig. Prediction aligned error (PAE) score of the predicted structure of Gp5/M of EAV Bucyrus.**
(PDF)

**S4 Fig. Comparison of different AlphaFold 3 models of GP5/M of EAV.**
(PDF)

**S5 Fig. Comparison of the AlphaFold 2 and AlphaFold 3 models of the EAV GP5/M dimer.**
(PDF)

**S6 Fig. Comparison of the AlphaFold 2 and AlphaFold 3 models of the PRRSV-2 GP5/M dimer.**
(PDF)

**S7 Fig. Multiple sequence alignment of the ectodomains of Gp5 of EAV, PRRSV-1 and PRRSV-2.**
(PDF)

**S8 Fig. Conservation of amino acids in M.**
(PDF)

**S9 Fig. Conservation of amino acids in Gp5.**
(PDF)

**S10 Fig. Details of parts of the transmembrane regions of three alphafold3 models of GP5/M of EAV.**
(PDF)

**S11 Fig. Structure of Orf3a and M of SARS-CoV-2 in a membrane context.**
(PDF)

**S12 Fig. Amino acid sequence conservation between Gp5 and M of Arteriviruses and Orf3a and M of SARS-CoV-2.**
(PDF)

## Author contributions

**Conceptualization:** Michael Veit.

**Funding acquisition:** Michael Veit.

**Investigation:** Michael Veit, Anna Karolina Matczuk.

**Writing – original draft:** Michael Veit.

**Writing – review & editing:** Anna Karolina Matczuk.

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
