## [Decision Letter · Decision Letter 0]

8 Jan 2026

Dear Dr. Veit,

Thank you for submitting your manuscript to PLOS ONE. After careful consideration, we feel that it has merit but does not fully meet PLOS ONE’s publication criteria as it currently stands. Therefore, we invite you to submit a revised version of the manuscript that addresses the points raised during the review process.

We look forward to receiving your revised manuscript.

Kind regards,

Vishwanatha R. A. P. Reddy

Academic Editor

PLOS One

Journal Requirements:

Reviewers' comments:

Reviewer's Responses to Questions

**Comments to the Author**

1. Is the manuscript technically sound, and do the data support the conclusions?

Reviewer #1: Yes

Reviewer #2: Yes

Reviewer #3: Partly

Reviewer #4: Yes

Reviewer #5: Partly

Reviewer #6: Partly

2. Has the statistical analysis been performed appropriately and rigorously?

Reviewer #1: I Don't Know

Reviewer #2: N/A

Reviewer #3: Yes

Reviewer #4: N/A

Reviewer #5: Yes

Reviewer #6: N/A

3. Have the authors made all data underlying the findings in their manuscript fully available?

Reviewer #1: Yes

Reviewer #2: Yes

Reviewer #3: Yes

Reviewer #4: Yes

Reviewer #5: Yes

Reviewer #6: Yes

4. Is the manuscript presented in an intelligible fashion and written in standard English?

Reviewer #1: Yes

Reviewer #2: Yes

Reviewer #3: Yes

Reviewer #4: Yes

Reviewer #5: Yes

Reviewer #6: Yes

Reviewer #1: This is a review on the manuscript entitled “Computational Modelling of the Equine Arteritis Virus GP5/M Dimer: Structural Basis for Immune Evasion, Virulence and Virus Budding” by Michael Veit and Anna Karolina Matczuk. Equine arteritis virus (EAV) is an important viral pathogen in horses. The major viral envelop proteins GP5 and M form heterodimer that is critical for virus assembly and budding. These proteins contain immunogenic epitopes that involve in viral pathogenesis and host immune responses. In this study, the authors used the AI-driven tool AlphaFold3 to predict the 3D structure of the EAV GP5/M dimer and compared to its homolog in porcine reproductive and respiratory syndrome virus (PRRSV) and SARS-CoV-2. Their findings provide in-depth molecular insights into the structure-function of the GP5/M dimer and established a foundation for rational design of EAV vaccines. This is a well-written manuscript with detailed protein structure analysis. This reviewer only has a few of the following comments/suggestions for the authors to address.

1. Line 41-43, “Other relevant arteriviruses include lactate dehydrogenase–elevating virus (LDV) and simian hemorrhagic fever virus (SHFV), the latter contains viruses with zoonotic potential”: The arterivirus family has been expanded and reclassified into 6 subfamilies containing 23 species with a number of more recently identified members, many of which are originated from monkeys. This information needs to be updated with references in the manuscript.

2. Line 152-153, “Only one GP5 sequence is predicted to be cleaved between residues 16 and 17…..”: More information is needed for this specific GP5 sequence. Is there a deletion in GP5 sequence or a possible annotation/sequencing error?

3. Lines 185-188: The rainbow pLDDT figure is helpful. It will be better if the authors describe more precisely which transmembrane region and β-sheet exhibit the highest confidence.

4. Lines 248–249: The authors noted that the EAV GP5 ectodomain (89 aa) is longer than PRRSV GP5 (26 aa). It is better to further discuss the functional implications of this extended domain and how it may relate to biological differences between EAV and PRRSV.

5. Lines 394–396: Part of this paragraph should be moved to discussion section. The structural analysis suggests that protein surface/epitope exposure could influence immune recognition. It is better to provide a discussion of previous studies in cell culture or animal models to support this notion.

6. Line 460, “…. with orf3a (12/19”: This sentence does not seem to be completed. It should be written as “…..with SARS-CoV-2 orf3a (12%/19%).”

Reviewer #2: Dear authors, the work is extremely interesting, and you have achieved a very good result using computer programs. Regarding figure 4, point C, I suggest highlighting the epitopes denoted in orange, blue, black, and green, which are not clearly visible, so that they can be properly appreciated.

Reviewer #3: Comments to the Authors

Title: The title implies experimentally validated mechanisms (“structural basis for virus budding”) that are not demonstrated. A more conservative title reflecting the predictive nature of the study is recommended.

Introduction: While the study focuses on EAV, the Introduction places disproportionate emphasis on PRRSV GP5/M biology. This weakens the narrative focus and makes EAV appear primarily as a comparative extension. The Introduction should more clearly define the specific knowledge gap for EAV and justify the comparative framework.

Interpretation of Structural and Evolutionary Comparisons: The reported structural similarities between arterivirus GP5/M and SARS-CoV-2 M/ORF3a are interesting, but conclusions should be limited to conserved topology and fold. Functional or evolutionary equivalence cannot be inferred from AlphaFold3 predictions alone, particularly given their static nature.

Limitations: The study relies exclusively on computational modeling. Regions with low confidence, especially the endodomains, limit interpretation of cytoplasmic interactions. Claims related to virus budding and assembly should therefore be explicitly presented as speculative and hypothesis-generating.

Reviewer #4: Dear authors, the work is extremely interesting, and you have achieved a very good result using computer programs. Regarding figure 4, point C, I suggest highlighting the epitopes denoted in orange, blue, black, and green, which are not clearly visible, so that they can be properly appreciated.

Reviewer #5: This manuscript uses AlphaFold3 to model the EAV GP5/M heterodimer, compares it with PRRSV homologs, and maps neutralizing epitopes, N-glycosylation sites, and virulence/persistence mutations to propose mechanisms of immune evasion and morphogenesis. The topic is relevant and the structure-guided mapping is potentially useful, especially the discussion linking epitope variability with nearby glycans and membrane-proximal mutations with functional phenotypes.

The main weakness is model confidence: the reported best model has pTM=0.48 and low-confidence endodomain regions (pLDDT<50), so several long-range interpretations (subunit arrangement, glycan “shielding,” and broad cross-Nidovirales extrapolations) need stronger quantitative support and more conservative wording.

1. Lines 173-175: The manuscript should more explicitly bound which conclusions are supported by high-confidence regions versus low-confidence regions, given the reported pTM=0.48 and low pLDDT in the endodomain;the authors should test robustness across the five AF3 candidate models by reporting whether key features (apical epitope region, helix4 membrane-proximal region, and the proposed interface) are consistent across models (e.g., segment RMSD and interface-contact overlap), rather than basing interpretation on a single “best” model.

2. The Methods should specify AF3 server/version/date, whether MSA/templates were used, exact sequence boundaries (signal peptide removal and any truncations), and the criteria for selecting the final model beyond global pTM; for a membrane-protein heterodimer, interface confidence is central, so the authors should report interface-relevant confidence (interface PAE or per-residue confidence at interface) and provide the final PDBs and basic visualization/analysis steps as supplementary materials.

3. Lines 500-505: As mapping epitope B/C variability near N38/N63 and the outbreak-associated N55 glycosylation, the authors should add quantitative exposure metrics (SASA of epitope residues, and occlusion estimates after adding simplified glycans), and either include a formal selection analysis (dN/dS/site tests) or soften “positive selection” language to “consistent with immune pressure,” since variability alone is not decisive.

4. Lines 383-396: In Fig 5, for virulence/persistence-associated mutations clustered near helix4 of Pro80/Ser82/Asp86 and related sites, the structural interpretation should include alternative mechanisms (notably indirect effects via glycosylation changes such as loss of the N63 site) and propose concrete validation experiments. GP5/M interaction assays with GP2/3/4, VLP budding/assembly assays, or targeted interface mutagenesis, to support the “interaction surface” hypothesis rather than presenting it as a primary explanation.

Reviewer #6: The paper presented by Veit and Matczuk is an extension of their previous in silico work performed on the PRRSV GP5/M heterodimer, by predicting the structure of the EAV GP5/M heterodimer and comparing its predicted structural features to PRRSV and coronavirus proteins. Whilst the work allows to speculate about structural conservation and the importance of certain domains as neutralising epitopes, no in vitro experimental evidence is provided to strengthen the findings, making it difficult to assess the relevance of the presented data. Therefore, this study is a nice piece of in silico modelling and prediction, which can provide the interested scientific community with structure prediction based ideas to improve our understanding of EAV biology and vaccine design.

Major comments:

Please be carefully with the phrasing you are using. You are not working with an experimentally determined structure, but with a structural model. Hence, I would kindly ask you to clearly indicate that you are working with an in silico generated model in the whole manuscript, including the title., e.g. ‘Computational modelling of the EAV GP5/M dimer suggests structural mechanisms of …’

Did you also recalculate the structure of the PRRSV GP5/M dimer using Alp

haFold3 to assess the structural similarity between AlphaFold2 and 3 predictions? I think this would be an important and interesting addition, or it should at least be discussed why this was not done.

Figures in general:

Please adept the colours in the figures used to distinguish the M and the GP5 chain. They are too similar for easy distinction (e.g. green and pale blue).

Please add a scale of the color gradient used whenever you depict the pLDDT. Please also consider to use the ‘usually’ used Alphafold output colour code, which is blue for high confidence, and red for low confidence.

Minor points:

Line 37 and 47 are repetitive. I think it would be important to indicate somewhere in the abstract that GP5/M are however not relevant for PRRSV neutralisation.

Line 48: Please give references and numbers for the significant economic losses.

Line 62: This reviewer is not sure such a statement is correct given which information is currently available in the literature, the statement should be weakened…

Line 149-151: Please indicate the tool used.

Line 156: What is ‘web logo’? E.g. please introduce the tool here.

Lines 214-216: the only reliable part of the prediction are the TMs

Lines 225-227: the depth of the multiple sequence alignment used by AlphaFold should give clues aout the prediction quality

Line 245: Can we really say it is high confidence if the pLDDT values are <70? This reviewer has some doubts.

Line 250: the proteins are homologous, allowing to compute a multiple sequence alignment, they do not "reveal limited homology" but reveal few conserved residues and an insertion in EAV GP5.

Line 269-273: a detailed description of the moderate confidence predicted EAV GP5 ectodomain is not really necessary.

Line 283: Single residue epitopes? I don’t think they exist… How were these ‘epitopes’ identified? How were the antibodies generated?

Line 315ff: How many sequences were available for these predictions and how do they represent the variability of EAV field strains? Please state here in the text.

Line 316: WebLogo is more accurately described in legend of Figure 4

Line 352f: What is the likelihood that an amino acid exchange from isoleucine to valine will cause significant biological property changes of the protein?

Figure 5C-E: please show the predictions in the context of the full protein. Are they compatible with the overall structure of the heterodimer?

Line 438ff: How can you be sure that this is structural cohesion and not a prediction bias?

Paragraph on structural homology with coronavirus proteins: Please include structure overlays as a figure to visualise your findings.

Line 452: the highest sequence conservation

Line 454: How was the homology relationship between Orf3 and GP5/M demonstrated ?

Line 475: if there was high sequence identity we could say close homologs.In the present case we could have (very) remote homologs, only if the homology relationship has been demonstrated.

Line 506ff: Given the high sequence variability of epitopes B and C, how can they be useful for vaccine antigen design? Will a mosaic design really be sufficient? And how should it be administered?

Line 517ff: Couldn’t this also imply that these sites need to be more conserved due to functional constraints? Isn’t it in the interest of the virus to have functionally important epitopes being less dominant? What about broadly neutralising epitopes?

**Do you want your identity to be public for this peer review?** For information about this choice, including consent withdrawal, please see our Privacy Policy

Reviewer #1: No

Reviewer #2: **Yes:** María Gabriela Echeverría

Reviewer #3: No

Reviewer #4: **Yes:** María Gabriela Echeverría

Reviewer #5: No

Reviewer #6: No

---

## [Author Response · Author response to Decision Letter 1]

17 Feb 2026

Point-by-point reply

Reviewer #1: This is a review on the manuscript entitled “Computational Modelling of the Equine Arteritis Virus GP5/M Dimer: Structural Basis for Immune Evasion, Virulence and Virus Budding” by Michael Veit and Anna Karolina Matczuk. Equine arteritis virus (EAV) is an important viral pathogen in horses. The major viral envelop proteins GP5 and M form heterodimer that is critical for virus assembly and budding. These proteins contain immunogenic epitopes that involve in viral pathogenesis and host immune responses. In this study, the authors used the AI-driven tool AlphaFold3 to predict the 3D structure of the EAV GP5/M dimer and compared to its homolog in porcine reproductive and respiratory syndrome virus (PRRSV) and SARS-CoV-2. Their findings provide in-depth molecular insights into the structure-function of the GP5/M dimer and established a foundation for rational design of EAV vaccines. This is a well-written manuscript with detailed protein structure analysis. This reviewer only has a few of the following comments/suggestions for the authors to address.

1. Line 41-43, “Other relevant arteriviruses include lactate dehydrogenase–elevating virus (LDV) and simian hemorrhagic fever virus (SHFV), the latter contains viruses with zoonotic potential”: The arterivirus family has been expanded and reclassified into 6 subfamilies containing 23 species with a number of more recently identified members, many of which are originated from monkeys. This information needs to be updated with references in the manuscript.

The information was updated:

Line 38-40: “Alphaarterivirus equid, common name equine arteritis virus (EAV), which causes respiratory and reproductive disease in horses and donkeys, is an enveloped plus-strand RNA virus in the Arteriviridae family, which comprises 6 subfamilies.”

Line 44-46: “Other relevant arteriviruses include lactate dehydrogenase–elevating virus (LDV) and many species of subfamily Simarterivirinae including simian hemorrhagic fever virus (SHFV), which infect non-humane primates and possess zoonotic potential (2-5).”

2. Line 152-153, “Only one GP5 sequence is predicted to be cleaved between residues 16 and 17…..”: More information is needed for this specific GP5 sequence. Is there a deletion in GP5 sequence or a possible annotation/sequencing error?

We have added the following information to the legend of Figure S1 rather than to the main text, in order to avoid overloading the manuscript with technical detail:

“The arrow marks the signal peptide cleavage site predicted by SignalP 5.0, located between residues 18 and 19 in all GP5 sequences except one. Amino acids at the −1 and −3 positions that promote signal peptide cleavage are highlighted in red. In the exceptional GP5 variant (NCBI Protein accession ABL14296.1), the small residue Ala at the −1 position is replaced by the bulky residue Asp, which prevents cleavage at this canonical site. Instead, SignalP predicts an alternative cleavage between residues 16 and 17 for this strain, accompanied by a marked reduction in the overall cleavage probability (from 0.9 to 0.4).”

3. Lines 185-188: The rainbow pLDDT figure is helpful. It will be better if the authors describe more precisely which transmembrane region and β-sheet exhibit the highest confidence.

We added the following information to this sentence (lines 193-195):

“This representation highlights that the membrane spanning parts of all six transmembrane regions and a prominent β-sheet and the following helix 1 within the GP5 ectodomain are predicted with the highest confidence, underscoring their structural reliability (Fig 1A, B, 2A).”

4. Lines 248–249: The authors noted that the EAV GP5 ectodomain (89 aa) is longer than PRRSV GP5 (26 aa). It is better to further discuss the functional implications of this extended domain and how it may relate to biological differences between EAV and PRRSV.

We added the following sentence to this part of the manuscript (lines 293-295): “The extended GP5 ectodomain of EAV may influence receptor engagement and other host‑factor interactions, which aligns with the broader cell and tissue tropism of EAV.”

5. Lines 394–396: Part of this paragraph should be moved to discussion section. The structural analysis suggests that protein surface/epitope exposure could influence immune recognition. It is better to provide a discussion of previous studies in cell culture or animal models to support this notion.

We have removed the final sentence of this paragraph: “This structural rearrangement likely repositions amino acids locally, potentially exposing new residues on the molecule’s surface.”. We think that a more extensive treatment of its immunological implications would go beyond the scope of the present manuscript

6. Line 460, “…. with orf3a (12/19”: This sentence does not seem to be completed. It should be written as “…..with SARS-CoV-2 orf3a (12%/19%).”

The sentence was corrected accordingly.

Reviewer #2: Dear authors, the work is extremely interesting, and you have achieved a very good result using computer programs. Regarding figure 4, point C, I suggest highlighting the epitopes denoted in orange, blue, black, and green, which are not clearly visible, so that they can be properly appreciated.

We thank the referee very much for the positive and encouraging comment. In the revised figure 4c, we have now labelled the four epitopes in the protein sequence with the letters A–D so that they clearly indicate the color‑coded epitopes shown in the structural representation.

Reviewer #3: Comments to the Authors

Title: The title implies experimentally validated mechanisms (“structural basis for virus budding”) that are not demonstrated. A more conservative title reflecting the predictive nature

We agree with this concern and deleted “virus budding” from the title and also replaced “structural basis” with “implications: “Computational Modelling of the Equine Arteritis Virus GP5/M Dimer: Implications for Immune Evasion and Virulence and Virus Budding”

Introduction: While the study focuses on EAV, the Introduction places disproportionate emphasis on PRRSV GP5/M biology. This weakens the narrative focus and makes EAV appear primarily as a comparative extension. The Introduction should more clearly define the specific knowledge gap for EAV and justify the comparative framework.

We thank the referee for this thoughtful comment. We carefully reviewed the Introduction in light of this suggestion. While we appreciate the concern about balance, we respectfully note that the Introduction is structured primarily around EAV. After the general overview of the Arteriviridae family (lines 38–49), the subsequent sections on pathogenesis (lines 50–57), vaccines (lines 58–66), viral receptors (lines 81–92), and antibody epitopes (lines 118–123) focus exclusively on EAV. The section on GP5/M processing (lines 98–117) includes selected PRRSV studies because most mechanistic insights available in the literature stem from PRRSV—particularly PRRSV‑2 strain VR‑2332—and these basic processes are considered highly conserved between PRRSV and EAV. To ensure completeness, we added an additional EAV‑specific reference and now cite all published studies on GP5/M processing in EAV that we are aware of.

The specific knowledge gap addressed in this study is defined later in the Introduction. In line 123, we state: “Despite its critical role in virus replication and immune evasion, the three-dimensional structure of the GP5/M complex remains unknown.”

Furthermore, lines 140–142 outline the aim of the study: “In this study, we employed AlphaFold3 to predict the GP5/M structure of EAV, focusing on known antibody epitopes and virulence factors. Knowing the precise 3D structure of an antigenic protein enables researchers to identify surface-exposed epitopes.”

Interpretation of Structural and Evolutionary Comparisons: The reported structural similarities between arterivirus GP5/M and SARS-CoV-2 M/ORF3a are interesting, but conclusions should be limited to conserved topology and fold. Functional or evolutionary equivalence cannot be inferred from AlphaFold3 predictions alone, particularly given their static nature.

AND

Limitations: The study relies exclusively on computational modeling. Regions with low confidence, especially the endodomains, limit interpretation of cytoplasmic interactions. Claims related to virus budding and assembly should therefore be explicitly presented as speculative and hypothesis-generating.

Thank you for these valuable comments. We have revised and condensed the summary at the end of the discussion to clearly acknowledge the limitations of an exclusively computational approach, including the low‑confidence regions in the cytoplasmic endodomains. The text now explicitly states that AlphaFold3 predictions cannot support functional, mechanistic, or evolutionary equivalence and that any interpretations related to cytoplasmic interactions, budding, or assembly are speculative. (lines 632-644).

“In summary, this study demonstrates that the AlphaFold3-predicted structure of the EAV GP5/M dimer reliably captures the architecture of parts of the ectodomain and transmembrane regions, offering new molecular insights into its organization. The model enables precise mapping of virulence-associated determinants and known antibody epitopes, which is consistent with a dual immune evasion strategy that combines antigenic drift with glycan shielding. At the same time, AlphaFold3 predictions reflect conserved topology and overall fold but cannot establish functional or evolutionary relationships. Because the models are computational and static, and several regions, particularly the cytoplasmic endodomains, are predicted with low confidence, functional or mechanistic equivalence cannot be inferred. Within these limitations, the conserved architecture of the transmembrane domains of Arterivirus GP5/M and those of SARS‑CoV‑2 M and ORF3a is consistent with, but does not demonstrate, the possibility of a distant shared ancestry or common principles of membrane‑associated assembly within the Nidovirales.”

Reviewer #4: Dear authors, the work is extremely interesting, and you have achieved a very good result using computer programs. Regarding figure 4, point C, I suggest highlighting the epitopes denoted in orange, blue, black, and green, which are not clearly visible, so that they can be properly appreciated.

We thank the referee very much for the positive and encouraging comment. In the revised figure 4c, we have now labelled the four epitopes above the protein sequence with the letters A–D so that they clearly indicate the color‑coded epitopes shown in the structural representation.

Reviewer #5: This manuscript uses AlphaFold3 to model the EAV GP5/M heterodimer, compares it with PRRSV homologs, and maps neutralizing epitopes, N-glycosylation sites, and virulence/persistence mutations to propose mechanisms of immune evasion and morphogenesis. The topic is relevant and the structure-guided mapping is potentially useful, especially the discussion linking epitope variability with nearby glycans and membrane-proximal mutations with functional phenotypes.

The main weakness is model confidence: the reported best model has pTM=0.48 and low-confidence endodomain regions (pLDDT<50), so several long-range interpretations (subunit arrangement, glycan “shielding,” and broad cross-Nidovirales extrapolations) need stronger quantitative support and more conservative wording.

1. Lines 173-175: The manuscript should more explicitly bound which conclusions are supported by high-confidence regions versus low-confidence regions, given the reported pTM=0.48 and low pLDDT in the endodomain; the authors should test robustness across the five AF3 candidate models by reporting whether key features (apical epitope region, helix4 membrane-proximal region, and the proposed interface) are consistent across models (e.g., segment RMSD and interface-contact overlap), rather than basing interpretation on a single “best” model.

We thank the reviewer for raising this important point. In response, we performed an additional AlphaFold3 prediction and compared two models from the initial run (the highest‑ and lowest‑confidence models) with one independently generated model from a second run. AlphaFold3 generates each model by iteratively denoising a cloud of random initial atomic positions, with each model initialized by a unique Seed. This allows us to assess the robustness of structural features across independently seeded predictions.

We have added the following text to the manuscript (lines 231-251): “We then compared the models with the highest and lowest confidence scores and performed an additional AlphaFold3 run to ensure convergence. Unlike deterministic calculations, AlphaFold3 builds structures by iteratively 'denoising' a cloud of random initial atomic positions. This process is governed by a 'Seed'—a starting number for the random number generator. Since each model, whether from the same or different runs, is initialized with a unique Seed, the algorithm explores a slightly different pathway during structure construction. This ensemble approach allows to further assess the reliability of the model. Consistent results across seeds indicate a robust, well-defined structure, while variability highlights regions of either physical flexibility or uncertainty in the prediction.

All three models exhibit the same set of structural elements in the same order, with the only notable exception being the M-protein N-terminus, which forms a short α-helix in one model but remains unstructured in others. Mapping of pLDDT scores confirms that the six transmembrane regions, the β-sheet, and helix 3 of the GP5 ectodomain are predicted with high confidence and align closely across all models. A minor misalignment is present in helix 4 across all models. A more pronounced deviation is observed between helices 1 and 2 when comparing the models from the two runs. Despite these localized variations in flexible loops and the N-terminus, the high structural convergence and confidence scores across the transmembrane core and key ectodomain elements demonstrate that the predicted heterodimer is robust.”

We also expanded the Results section to clarify which conclusions are supported by high‑confidence regions and to indicate where the models show consistent structural features.

After the section on epitope localization, we added (lines 338-340): “Note that the β-sheet and the following helix 3 are predicted in all models with the highest accuracy and their structures align well.”

After the section on virulence determinants, we added (lines 446-448): “Note that the spatial position of helix 4 varies among the AlphaFold models, but it consistently remains connected to helix 3 by a loop, and helix 3 itself aligns very well across all models (S4 Fig.).”

Finally, in the paragraph describing the transmembrane regions, we added (lines 483-485):

“Interestingly, all three models predict identical electrostatic interactions between the transmembrane segments of GP5 and M, indicating that the prediction of the interface is robust (S10 Fig).”

2. The Methods should specify AF3 server/version/date, whether MSA/templates were used, exact sequence boundaries (signal peptide removal and any truncations), and the criteria for selecting the final model beyond global pTM; for a membrane-protein heterodimer, interface confidence is central, so the authors should report interface-relevant confidence (interface PAE or per-residue confidence at interface) and provide the final PDBs and basic visualization/analysis steps as supplementary materials.

This part of the Methods section (line 647) was substantially expanded and now contains the requested information.

3. Lines 500-505: As mapping epitope B/C variability near N38/N63 and the outbreak-associated N55 glycosylation, the authors should add quantitative exposure metrics (SASA of epitope residues, and occlusion estimates after adding simplified glycans), and either include a formal selection analysis (dN/dS/site tests) or soften “positive selection” language to “consistent with immune pressure,” since variability alone is not decisive.

We thank the referee for this suggestion

---

## [Editor Report · Decision Letter 1]

19 Feb 2026

Computational Modelling of the Equine Arteritis Virus GP5/M Dimer: Implications for Immune Evasion and Virulence

PONE-D-25-56108R1

Dear Dr. Veit,

We’re pleased to inform you that your manuscript has been judged scientifically suitable for publication and will be formally accepted for publication once it meets all outstanding technical requirements.

Kind regards,

Vishwanatha R. A. P. Reddy

Academic Editor

PLOS One
---

## [Editor Report · Acceptance letter]

PONE-D-25-56108R1

PLOS One

Dear Dr. Veit,

I'm pleased to inform you that your manuscript has been deemed suitable for publication in PLOS One. Congratulations! Your manuscript is now being handed over to our production team.

Kind regards,

on behalf of

Dr. Vishwanatha R. A. P. Reddy

Academic Editor

PLOS One